# Whole Genome Sequencing Characterization of HEV3-*e* and HEV3-*f* Subtypes among the Wild Boar Population in the Abruzzo Region, Italy: First Report

**DOI:** 10.3390/microorganisms8091393

**Published:** 2020-09-11

**Authors:** Giuseppe Aprea, Silvia Scattolini, Daniela D’Angelantonio, Alexandra Chiaverini, Valeria Di Lollo, Sabrina Olivieri, Maurilia Marcacci, Iolanda Mangone, Stefania Salucci, Salvatore Antoci, Cesare Cammà, Adriano Di Pasquale, Giacomo Migliorati, Francesco Pomilio

**Affiliations:** 1Istituto Zooprofilattico Sperimentale dell’Abruzzo e del Molise “G. Caporale”, 64100 Teramo, Italy; d.dangelantonio@izs.it (D.D.); a.chiaverini@izs.it (A.C.); v.dilollo@izs.it (V.D.L.); s.olivieri@izs.it (S.O.); m.marcacci@izs.it (M.M.); i.mangone@izs.it (I.M.); s.salucci@izs.it (S.S.); c.camma@izs.it (C.C.); a.dipasquale@izs.it (A.D.P.); g.migliorati@izs.it (G.M.); f.pomilio@izs.it (F.P.); 2Local health unit “ASL Teramo”, 64100 Teramo, Italy; salvatore.antoci@aslteramo.it

**Keywords:** hepatitis E (HEV) virus, public health, whole genome sequence (WGS), wild boar

## Abstract

Hepatitis E virus (HEV) is an emergent zoonotic pathogen, causing worldwide acute and chronic hepatitis in humans. HEV comprises eight genotypes and several subtypes. HEV genotypes 3 and 4 (HEV3 and HEV4) are zoonotic. In Italy, the most part of HEV infections (80%) is due to autochthonous HEV3 circulation of the virus, and the key role played by wild animals is generally accepted. Abruzzo is an Italian region officially considered an HEV “hot spot”, with subtype HEV3-*c* being up to now the only one reported among wild boars. During the year 2018–2019, a group of wild boars in Abruzzo were screened for HEV; positive RNA liver samples were subjected to HEV characterization by using the whole genome sequencing (WGS) approach methodology. This represents the first report about the detection of HEV-3 subtypes *e* and *f* in the wild boar population in this area. Since in Italy human infections from HEV 3-*e* and *f* have been associated with pork meat consumption, our findings deserve more in-depth analysis with the aim of evaluating any potential correlation between wild animals, the pork chain production and HEV human infections.

## 1. Introduction

With more than 21,000 human cases over the last 10 years [1], hepatitis E virus (HEV) is the most common cause of acute viral hepatitis worldwide. The virus is principally transmitted by the fecal–oral route, and it is responsible, in humans, for a generally self-limiting acute hepatitis that could evolve into fatal hepatitis in pregnant women [2], or into chronic liver infections in immunocompromised individuals [3]. HEV is a member of the family *Hepeviridae*, genus *Orthohepevirus*, species *Orthohepevirus A*. This species comprises eight genotypes (HEV1–8) [4] and several subtypes (n. 31; update at 18 August 2020 [5]). In developing countries, genotypes 1 and 2 (HEV1 and HEV2) predominate, causing large epidemics by waterborne transmission [6]. Instead, in industrialized countries, the infection is mainly associated with travelling to endemic areas or it has an autochthonous origin [7], predominantly involving genotypes 3 and 4. 

While HEV1 and HEV2 are restricted to humans, HEV3 and HEV4 are zoonotic and potentially foodborne transmitted, infecting several animal species, including those used in food production, mainly domestic and wild ungulates (e.g., boars and deer [8]). 

Furthermore, in Italy, a survey from 2012–2016, highlighted the high prevalence of autochthonous HEV3 in human infections from consumption of raw/undercooked pork sausages. Moreover, the key role of wild boars as natural reservoirs of the virus is also recognized [9]. 

Data on HEV subtypes from wild animals reported in Italy are very heterogeneous. In particular, HEV 3-*e*, 3-*f* and 3-*c* have been reported in wild boars in northwestern Italy [10,11]. Moreover, HEV 3-*a*, 3-*c*, 3-*e*, 3-*f*, 3-*l* and 3-*j* are the most frequently reported in Central west of Italy [12,13], while 3*j* was also mentioned in South Italy [14]. Abruzzo is a region in the central east part of Italy, and it is nationally recognized as an HEV “hot spot” because of the high concentration of HEV human cases and seroprevalence among blood donors [15]. To date, only HEV3-*c* has been reported among wild boars in this region [11,14,16,17] and, apart from pork meat and liver consumption, in Abruzzo, there is also a very common habit to use wild boar meat and entrails for direct human consumption or to produce sausages and salami [14]. On the other hand, HEV3-*e* and 3-*f* are most frequently reported in Italy as cause of human infections from the consumption of raw/undercooked pork meat [18,19]. In addition, these genotypes are the most frequently reported among pigs [14,18]. In this paper, we describe a study on HEV detection in liver samples from a group of 431 wild boars hunted in the Abruzzo region from October 2018 to November 2019. In particular, liver samples were processed for the detection of HEV RNA by real-time RT-PCR; positive samples were subjected to whole genome sequencing (WGS) and phylogenetic analysis by using a distinct set of whole genome reference sequences [5,20,21] that have been recently proposed by the European Food Safety Authority (EFSA), and which should be strictly used in the future for sequence analysis [1,5]. 

Our results showed for the first time evidence of the circulation of two HEV3-*e* strains and one HEV3-*f* strain from wild boars in the Abruzzo region. This finding poses a significant point for discussions in order to evaluate the potential correlation existing between human cases of hepatitis E and HEV in pigs and wild boars in this region. Above all, the possible cross transmission of HEV from the wild boar population to the pig industry and vice versa should be investigated. Moreover, data from this work enforce the still small amount of HEV–WGS information available in literature, which is crucial for any kind of further speculations in relation to HEV foodborne transmission and source attribution [1].

## 2. Materials and Methods 

### 2.1. Sample Processing and RNA Extraction 

During the period of October 2018–November 2019, liver samples were collected from individual wild boars (Sus scrofa scrofa) hunted in Abruzzo and uniformly distributed in the region. For the viral extraction step, approximately 1 ± 0.2 g of liver sample was added to 1 ± 0.2 g of quartz fine granules (Merck, Darmstadt, Germany) and to 5 ± 0.2 mL of pH 7.2 phosphate buffered saline (PBS). After homogenization, each sample was clarified by centrifugation at 1500× *g* for 2 min at room temperature (18–22 °C). Than supernatant was collected, and 500 µL of it was processed for viral RNA extraction by using the Nuclisens magnetic extraction Kit (Biomerieux, Marcy-l’Etoile, France) and the Nuclisens MiniMag platform, according to the manufacturer’s instructions.

### 2.2. Real-Time RT–PCR Assay

A real-time reverse transcriptase polymerase chain reaction (RT-PCR) assay targeting the open reading frame (ORF) 3 region of HEV was used to amplify RNA samples. Two different RNA sample concentrations were tested: undiluted and diluted (1/10; for reduction of any potential sample inhibitory effect expressed during the real-time PCR amplification step [22]). The RT PCR was performed using UltraSense^TM^ One-Step Quantitative RT-PCR System kit (Invitrogen) and the following primers and probes: forward primer HEV-F (5′-GGTGGTTTCTGGGGTGAC-3′), reverse primer HEV-R (5′-AGGGGTTGGTTGGATGAA-3′) and HEV probe (TaqMan probe) HEV-P (5′-FAM-TGATTCTCAGCCCTTCGC-BGQ1-3′) [23]. An internal amplification control (IAC) was also included in each reaction, using a minor groove binder (MGB) TaqMan probe: IACP (5′-VIC-CCATACACATAGGTCAGG-MGB-NFQ-3′) [24]. Non-template control (NTC) as well as viral RNA derived from HEV positive control strain (Friedrich-Loeffler-Institut, Federal Research Institute for Animal Health, Greifswald, Germany) were also tested in each run. All the PCR analyses were performed by 7500 Fast Real-Time PCR System (Applied Biosystems, Carlsbad, CA, USA) with the use of the following thermal profile: 50 °C for 15 min, 95 °C for 2 min, 45 cycles of 95 °C for 10 s, 55 °C for 20 s and 72 °C for 15 s. The reaction mix (20 μL) contained 10 μL of sample RNA, 1× RNA UltraSense master mix (RNA UltraSense One-Step Quantitative RT–PCR System, Thermo Fisher Scientific, Rodano, Italy), 20× RNA UltraSense Enzyme Mix, 250 nM each primer (Tema Ricerca, Castesano, Italy), 100 nM probe (Thermo Fisher Scientific), 0.6 μL of IAC and 50 nM IAC probe [25]. The samples were considered positive when they showed a cycle threshold (Ct) value < 40. Samples with Ct ≥ 40 were deemed negative or inhibited depending on the result for IAC. The sample was considered negative if the Ct value for IAC was similar to the Ct value for IAC in the negative template control (NTC). If the sample IAC Ct value was higher than the NTC IAC Ct value, the sample was considered inhibited, and an additional 10-fold dilution of the extracted RNA was tested [22].

### 2.3. Sequencing

RNA from the samples that tested positive and showed a Ct value < 31 on real-time RT–PCR was analyzed by WGS and phylogenetic analysis. Total RNA was treated with TURBO DNase (Thermo Fisher Scientific, Waltham, MA, USA) at 37 °C for 20 min and then purified by RNA Clean and Concentrator-5 Kit (Zymo Research, Irvine, CA, USA). RNA was used for the assessment of sequencing independent single primer amplification protocol (SISPA) using SuperScript^®^ IV Reverse Transcriptase (Thermo Fisher Scientific, Waltham, MA) and a combination of two primers including the random-tagged primer FR26RV-N 5′-GCCGGAGCTCTGCAGATATCNNNNNN-3′ and the poly-A tagged primer FR40RV-T 5′-GCCGGAGCTCTGCAGATATCTTTTTTTTTTTTTTTTTTTT-3′ [26]. The PCR product was purified by ExpinTM PCR SV (GeneAll Biotechnology CO., Seoul, Korea) and then quantified using the QuantiFluor One ds DNA System kit (Promega, Madison, WT, USA). Libraries were prepared using Nextera DNA Flex Library Prep (Illumina Inc., San Diego, CA, USA) according to the manufacturer’s protocol. Deep sequencing was performed on the MiniSeq (Illumina Inc., San Diego, CA, USA) by the MiniSeq Mid Output Kit (300-cycles) and standard 150 bp paired-end reads.

### 2.4. Phylogenetic Analysis

For the analysis of WGS data, an in-house pipeline was used [27] which included steps for trimming (trimmomatic v0.36) and quality control check of the reads (FastQC v0.11.5). Genome de novo assembly of paired-end reads was performed using SPAdes v3.11.1 [28]. Based on genome assemblies, a de novo filtering for the scaffolds with a minimum length of 200 nucleotides and a matching for the best reference for each assembly using ABRicate was carried out [29]. Finally, a mapping with the references found in the previous step using Bowtie2 (v.2.1.0) was performed [30]. The generated draft genomes of 9/12 HEV RNA samples were submitted to the National Center for Biotechnology Information (NCBI) under the following accession numbers: MT840360 (2018.AZ.6050.8.7), MT840361 (2018.AZ.6050.8.10), MT840362 (2018.AZ.6050.8.11), MT840363 (2019.AZ.4671.2.1), MT840364 (2019.AZ.5758.1.4), MT840365 (2019.AZ.5758.1.5), MT840366 (2019.AZ.6234.1.1), MT840367 (2019.AZ.6360.1.2). The sequence 2018.AZ.6050.8.2 will be available from 1 December 2021 under the accession number MT840359. 

In order to visualize the worldwide phylogenetic position of HEV samples of this study, a total of 82 HEV complete genome sequences were selected and downloaded from NCBI, chosen among those suggested by EFSA as the ones to be strictly used for these kinds of WGS phylogenetic studies [1]. In particular, WGS from Smith et al. (2016, 2020) [5,20] and Rodriguez et al. (2015) [21] were primarily taken into account. 

The multi-alignment of overall 94 sequences was carried out with Clustal Omega, and we built the maximum likelihood tree (ML) using the Tamurai–Nei model [31], and the 500 replicates bootstrap method using MEGAX software.

In order to validate our phylogenetic results, we verified the subtype assignment of the 12 draft genomes by using the HEVnet tool (Available online: https://www.rivm.nl/mpf/typingtool/hev/) [32].

## 3. Results

During the period of October 2018–November 2019, 431 liver samples from wild boars uniformly hunted within the region Abruzzo were analyzed. Overall, 39 samples tested positive (*P* = 9.05%) with Ct values that ranged from 21 to 38.2. Twelve RNA samples, with a Ct value < 31, gave a valid result after WGS analysis and genotyping. The phylogenetic tree (Figure 1) showed that all viral isolates belonged to HEV-3. In particular, 2/12 belonged to subtype *e* and 1/12 to subtype *f*. The other 9 viral isolates, instead, belonged to HEV3 subtype *c* (Table 1). Table 1 also shows hunting areas and wild boars’ ages and sex.

The sequencing returned a number of reads ranging from 20,142 to 5,070,316 among the samples, with a Q30 ratio higher than 84% (Table 2). After the quality check and the trimming, the number of clean reads ranged from 14,560 to 3,686,844 for each sample. The ratio of Q30 bases comprised more than 92% clean data (Table 2). The mean length sequence of the 12 assemblies after the mapping was 7079 (min: 6736; max: 7210), with mean %GC of 55.5 (min: 55.2; max: 56). Four draft genomes (2018.AZ.6050.8.6, 2019.AZ.6232.1.6, 2019.AZ.6232.1.4, 2019.AZ.5758.1.5) showed a very low coverage against the relative reference genomes as indicated by ABRicate. 

The authors decided to submit to NCBI 9/12 draft full genomes because 3/12 (2019.AZ.6232.1.6, 2019.AZ.6232.1.4, 2018.AZ.6050.8.6) showed large gaps within the ORF 1 region. Nevertheless, the phylogenetic tree was built by using all the 12 HEV full genomes that fell into their relative clusters as shown in Figure 1. The HEVNet tool also confirmed the subtyping results for all the draft sequences apart from 2019.AZ.5758.1.5, which was identified as an HEV3 “presumptive” *l* subtype. 

## 4. Discussion

To the best of our knowledge, this is the first report about the circulation of HEV3 subtypes *e* and *f* in the wild boar population in Abruzzo. This region is of particularly importance in Italy since it is an “hot spot” of human HEV infection for the high seroprevalence of anti-HEV IgG [15]. Moreover, this work enforces the small number of HEV–WGS information available in the literature. 

A number of data from a small part of HEV genomes are available on HEV prevalence in pigs, wild boar and deer populations in Italy [33], and virus detection has been also demonstrated in slaughtered pigs, processed pork liver or meat products [19]. Nevertheless, there is still lack of data, and the link between HEV in animals and the source of infection in humans is not yet fully established. Many hypotheses associate infections from HEV3 subtypes *e* and *f* in humans to the consumption of raw or undercooked pork meats, and these findings suggest that the risk is also associated with the consumption of wild boar meat, or with the possibility that wild animals could infect pig herds because of weak biosecurity protocols. These data need further investigation, because there is little information of which most is associated with studies on partial HEV genome sequences. In fact, HEV genotype and subtype determination is still extensively performed by RT-PCR amplification of fragments of the HEV genome followed by sequencing and comparison with previously typed strains. Mainly ORF 1 and ORF 2 regions of the HEV genomes are investigated [1]. The sequence length used for typing largely influences the validity of the result. Vina-Rodriguez et al. (2015) [1,21] suggested the use of sequences with more than 1000 nucleotides for classification, whereas sequences shorter than 200 nucleotides should be avoided from subtyping because reduced sequence lengths lead to lower percentage bootstrap supports and are therefore often not significant. In a recent paper, Smith et al. (2020) proposed to admit for phylogenetic studies only those HEV sequences longer than 5000 nucleotides [5]. According to an opinion from EFSA, the use of whole genome sequences would clearly allow for the best characterization of a strain, including the unequivocal assignment to a specific subtype [1]. 

While the main problems related to the use of WGS involved the expensive cost of analysis, lately, the capability for efficient generation and analysis of whole genome sequence data is becoming more and more affordable, above all in industrialized countries. Another limitation in the WGS analysis is represented by the fact that, for some sample types, e.g., processed foods, a low amount of the HEV genome is recovered, and difficulties in its extraction and purification often restrict sequencing to short genome fragments [1]. From our experience, we were able to obtain good WGS results from RNA with a Ct value < 31; in our previous work, instead, we were able to perform successful ORF sequencing analysis starting from RNA viral isolates that were tested positive to a screening PCR with Ct < 34 [14]. For this reason, there is the need for developing techniques that enable more efficacious RNA recovery from samples in order to produce WGS data. Nevertheless, at this stage, scientists should be encouraged to try producing full viral genome sequences, at least from samples showing an HEV RNA Ct < 31 at the screening step. In fact, the use of WGS analysis is recommended for more precise data for trace back and forward analysis, and for evolution and source attribution studies, while the extensive use of phylogenetic studies by using only partially genome sequences are be not robust enough for such in-depth investigation. 

In relation to the results from this study, data continue to show, also for 2018 and 2019, the circulation in Abruzzo of HEV3-*c* strains (n. 9) among the wild boar population, as reported previously during 2015–2016 [14]. Though clustering together within the HEV3-*c* group, the strains detected in 2018 (n. 5, 89.5–99.9 % nucleotide identity) and the strains from samples of 2019 (n. 4, 76.9–96.6 % nucleotide identity) shared a nucleotide identity (n.i.) among the two groups ranging from 71.3% and 89.5%. This could be partially explained by potential genome mutations occurring in the strains over the year, but these data need further investigation. Moreover, the HEV3-*c* strains detected in Abruzzo in 2018, apart from sharing 86.8–95.8% n.i. with an HEV3-*c* strain detected in wild boars in Germany [34], confirming what was already found in 2016 [14], they also shared 86.4–96.6% n.i. with a Dutch human strain (MN614141.1) [35]. Interestingly, these two foreign HEV3-*c* strains shared a very high n.i. (96.6%), which could explain a potential correlation between wild boars and human infections. 

The two HEV3-*e* strains found in this study shared 97.5% n.i. In particular, they were found to be genetically related with other HEV3-*e* strains detected in France (JQ953665.1) [36] and UK pigs (MH184582.1, MH184584.1) [37], showing an overall n.i. of 88.1–89.7%. Interestingly, our HEV3-*e* strains also showed a n.i. of 89% with the strain AB248521.1 [5,20], which was also founded in Italy and detected from several hosts such as humans, pigs and wild boars. 

The HEV3-*f* strain identified in this study fell into the relative cluster (Figure 1), and it was genetically related to other European HEV3-*f* strains from humans (Germany, KJ873911.1) [38], wild boars (Sweden, KT581445.1, KT581446.1) [37], and pigs (Sweden EU360977.1 [39] and KT581447.1 [37]). Nevertheless, the results from the HEVNet tool characterized this strain as belonging to a presumptive *l* subtype (update at 18 August 2020). This hypothetical mismatch could be explained by the low coverage expressed after the mapping against the relative reference (KT447527, indicated by ABRicate), but could also support the methodology suggested by Smith et al. (2020) [5] that based the WGS data analysis on the exclusion of high variable regions or recombinant sequences from full draft genomes prior to their analysis. Furthermore, only Smith et al. (2020) [5] and De Sabato et al. (2018) [13] proposed full genome sequences related to the HEV3-*l* subtype in Italy, both showing 79.4% n.i. with our HEV3-*f* strain. For all these reasons, the authors believe that the HEV3-*f* full genome sequence from this study deserves further in-depth investigation. 

## 5. Conclusions

In Abruzzo, during October 2018–November 2019, the circulation of three HEV 3 subtypes were reported: HEV 3-*c*, 3-*e* and 3-*f*. In particular, this is the first report about HEV 3-*e* and *f* genotypes among the wild boar population in this Italian region which is officially known for counting a high number of human infections.

HEV 3-*e* and HEV 3-*f* strains are the main subtypes involved in human infections in Italy and are generally transmitted from pork meat. Our results obtained from WGS analysis performed with a set of full genome reference sequences, recommended at international level [1,5] and compared with other available WGS data, showed the existence of nucleotide sequence similarities between our strains and others deriving from wild boars, pigs and humans in Europe, above all from the North.

The authors highlight the importance of producing an increasing amount of WGS data, also in Italy, with the purpose of delving deeper into source attribution studies. In fact, to date, there is a very small amount of existing information on HEV from food and pork chain productions and most of it is limited to partial ORF genomic studies. Currently, it is impossible in Italy to compare WGS of wild boars to human and swine HEV3 sequences because of the lack of these WGS data. In-depth studies, above all in relation to source attribution investigation, will probably be feasible as soon as more full genome sequences are available. Moreover, the HEV3-*f* strain found in this study should be further investigated. 

Finally, the HEV sequences determined in this study may be useful for comparing present and future human isolates to study transmission events between wild boars, humans, and farmed pigs.

## Figures and Tables

**Figure 1 microorganisms-08-01393-f001:**
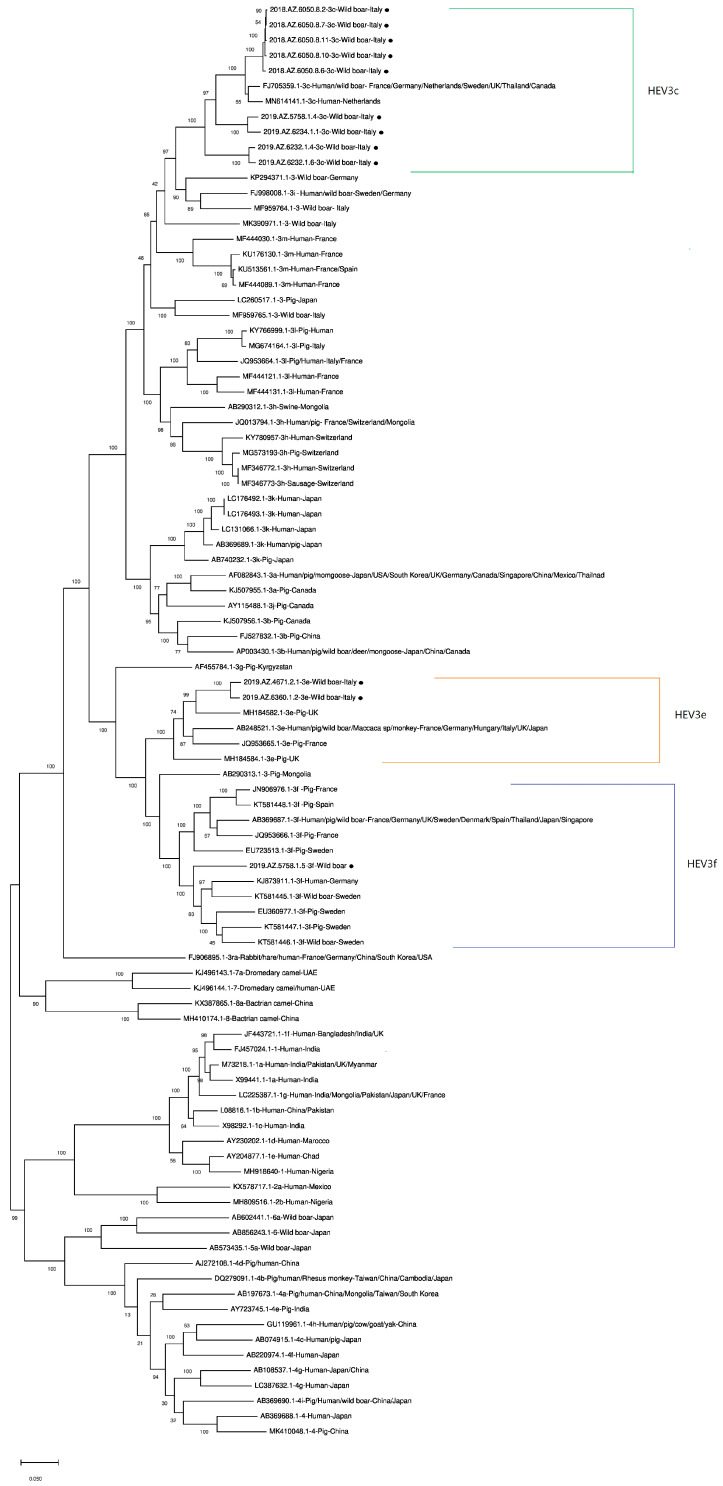
Phylogenetic tree based on HEV whole genome sequences (WGS). The tree was generated by neighbour joining using the Tamura 3-parameter model. Bootstraps values > 70 are reported, obtained by supplying statistical support with bootstrapping of 500 replicates. Sequences from this study are indicated by black circles.

**Table 1 microorganisms-08-01393-t001:** Whole genome sequencing (WGS) identification (ID), wild boar hunting area, year of hunting, age and sex of hunted wild boars and hepatitis E virus (HEV) subtypes.

ID	Hunting Area	Year	Age	Sex	HEV Subtype
2018-AZ-6050-8-2	L’Aquila	2018	6 months	Female	HEV3-*c*
2018-AZ-6050-8-6	L’Aquila	2018	6 months	Female	HEV3-*c*
2018-AZ-6050-8-7	L’Aquila	2018	6 months	Female	HEV3-*c*
2018-AZ-6050-8-10	L’Aquila	2018	6 months	Female	HEV3-*c*
2018-AZ-6050-8-11	L’Aquila	2018	10 months	Female	HEV3-*c*
2019-AZ-6360-1-2	Teramo	2019	12 months	Female	HEV3-*e*
2019-AZ-6234-1-1	L’Aquila	2019	6 months	Male	HEV3-*c*
2019-AZ-6232-1-6	L’Aquila	2019	-	Female	HEV3-*c*
2019-AZ-6232-1-4	L’Aquila	2019	-	Female	HEV3-*c*
2019-AZ-5758-1-5	Pescara	2019	-		HEV3 *f*
2019-AZ-5758-1-4	Pescara	2019	-		HEV3-*c*
2019-AZ-4671-2-1	L’Aquila	2019	6 months	Male	HEV3-*e*

**Table 2 microorganisms-08-01393-t002:** Data of WGS performance.

Sample_ID	Total Rawreads	Q30 Rawreads	Total Trimreads	Q30 Trimreads	Sequence Lenght	%GC	Coverage
2018.AZ.6050.8.2	5,070,316	84.71	3,686,844	92.72	7210	55.5	775.17
2018.AZ.6050.8.6	20,142	85.72	14,560	93.05	6736	55.4	5.87
2018.AZ.6050.8.7	2,483,214	84.95	1,705,664	93.09	7067	55.7	369.24
2018.AZ.6050.8.10	1,560,834	84.63	1,114,066	92.82	7006	55.6	88.12
2018.AZ.6050.8.11	924,066	85.86	832,502	93.11	7039	55.7	63.06
2019.AZ.4671.2.1	458,770	95.47	440,458	97.71	7237	55.1	1678.89
2019.AZ.5758.1.4	1,568,372	86.03	1,469,764	92.5	7107	56	54.62
2019.AZ.5758.1.5	767,482	93.91	730,209	97.93	7024	55.7	8.39
2019.AZ.6232.1.4	1,224,774	84.32	1,124,515	93.19	7042	55.3	8.58
2019.AZ.6232.1.6	488,730	92.35	451,677	97.72	7206	55.4	7.49
2019.AZ.6234.1.1	844,282	94.76	806,030	97.95	7087	55.6	174.86
2019.AZ.6360.1.2	444,820	90.58	423,140	97.63	7191	55.2	130.17

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
