# Peer review of "Whole Genome Sequencing Characterization of HEV3-e and HEV3-f Subtypes among the Wild Boar Population in the Abruzzo Region, Italy: First Report"

_microorganisms, 2020, doi:10.3390/microorganisms8091393_

Round 1
Reviewer 1 Report
Aprea et al. interested in the prevalence of HEV RNA in 431 wild boards living in Abruzzo, Italy. Liver samples collected between October 2018 and November 2019 were screened: 39 samples were positive. Samples with CT values<31 have been subjected to Whole Genome Sequencing. The authors determined the sequences for 12 samples. Most were HEV3c but one was proposed to belong to another new subtype l-p. The main issue of the paper is that it does not refer to the recent paper published by Smith et al. who proposed an update of the HEV classification, with the suggestion of new subtypes.
Major comments:
Figure 1: it not clear why the sequence 2019-AZ-5758-1-5 would belong to a new subtype since it looks as if it is in the cluster 3f. The reason why this sequence could belong to a new subtype should be indicated clearly in the result section. In addition, Smith et al. has recently published an update of the classification (Smith, J Gen Virol. 2020) and proposed 3k, 3l and 3m as new subtypes. Please add the corresponding sequences in the figure, this could help to determine the subtype of the sequence 2019-AZ-5758-1-5.
Minor comments:
In the introduction, the classification of HEV could be described a bit more. In particular, it could be mentioned that the 8 main genotypes belong to the species Orthohepevirus A.
Line 110, ref 22 : do the authors mean they adapted the protocol from one used for Sars-CoV2 ?
Figures, Tables and schemes do not desserve a dedicated section (3.1 Figures, Tables and schemes)
I suggest to delete the supplementary table 1 and instead put the informations in the figure. For instance, for AB074915.3, the figure could mention AB074915.3-4c-Human-Japan.
There are some typos in the manuscript: sousages instead of sausages…
Author Response
Dear Editor and Reviewers, this is Aprea Giuseppe, first author of the draft paper ID 903044, now entitled “Whole Genome Sequencing characterization of HEV3-e and HEV3-f subtypes among the wild boar population in Abruzzo region, Italy: first report”.
Together with all the co-authors, I would like to thank the reviewers for the very precious work done and for the advices provided that definitely improve the quality of our results and discussions. Unfortunately, because of the Covid emergence, that still involves most of our time and work activities, we missed the very recent paper from Smith et 2020, which we agree is fundamental for any actual studies that focus on HEV WGS analysis.
So told we restructured our paper updating and including the HEV genome references according to Smith et al, 2020. In particular, it is interesting to highlight that the HEV3-c and 3-e draft genome sequences still fell into the same groups as analysed in the previous version of our paper. Instead, from the new analysis, the sequence related to the whole genome HEV strain that was previously reported as a presumptive “l” subtype, now it clearly fall into the HEV3-f cluster. However, the low coverage of this sequence against the 3-f references and the results from HEVNet tool (still providing an hypothetical presumptive l subtype classification for this strain) suggest the existence of some peculiarities about the WGS 2019.AZ.5758.1.5 (MT840365), that should deserve further investigations, as we discussed in the paper.
Moreover, we edited our draft paper according to all the other revisions suggested by the reviewers, which I provide in a point by point reply, as it follows.
The edited parts inside the draft paper have been highlighted in yellow and signed in “Track changes”.
Replies to reviewer n. 1:
Major comments:
Figure 1: it not clear why the sequence 2019-AZ-5758-1-5 would belong to a new subtype since it looks as if it is in the cluster 3f. The reason why this sequence could belong to a new subtype should be indicated clearly in the result section. In addition, Smith et al. has recently published an update of the classification (Smith, J Gen Virol. 2020) and proposed 3k, 3l and 3m as new subtypes. Please add the corresponding sequences in the figure, this could help to determine the subtype of the sequence 2019-AZ-5758-1-5.
R: The Figure 1 has been replaced by an update version according to Smith et al.2020, including all the references suggested by those authors. Now, 2019-AZ-5758-1-5 clearly fall into the HEV3-f cluster
Minor comments:
In the introduction, the classification of HEV could be described a bit more. In particular, it could be mentioned that the 8 main genotypes belong to the species Orthohepevirus A.
R: Thanks for the comment. The HEV classification has been fully described in lines 37-39.
Line 110, ref 22 : do the authors mean they adapted the protocol from one used for Sars-CoV2 ?
R: yes, this is what the authors did
Figures, Tables and schemes do not deserve a dedicated section (3.1 Figures, Tables and schemes)
Re: This dedicated section has been removed
I suggest to delete the supplementary table 1 and instead put the informations in the figure. For instance, for AB074915.3, the figure could mention AB074915.3-4c-Human-Japan.
Re: Thanks for your advice. The supplementary table has been removed and the information about animal origin and State have been included in Fig.1 as suggested in the example
There are some typos in the manuscript: sousages instead of sausages…
Re: We are sorry for the mistakes. All the paper has been checked and the editing applied as all over highlighted in the draft paper.
Replies to reviewer n. 2:
The authors describe the results of a HEV screening exercise among wild boars in a region of Italy with high human hepatitis E prevalence. They find evidence of circulation of HEV-3c, 3e and what they describe to be a putative novel HEV-3 subtype. It is good that they do WGS for accurate genotyping.
Re: We want to thank the reviewer for sharing the same opinion. We are strongly convinced that the use of good WGS from HEV viral strains should be encouraged to be produced for any robust data analysis in the future.
The main problem is that Smith DB et al have just updated reference sequences for HEV in July 2020 (PMID: 32469300) and they have already assigned sequences to new HEV subtypes 3l and 3m. The rules for assigning new subtypes have been clearly described. The authors will need to update some of their WGS reference sequences and re-run the analysis with respect to this new publication. As it stands, their conclusion that they have a novel subtype has not been confirmed.
Re: We again apologize for not having come across this very important updates from Smith et al.2020. We re-analysed our WGS according to some of Smith et al. 2020 rules and by using all the references stated in his update version.
- Is consumption of wild boar meat common in the Abruzzo region?
Re: Yes, it is. Now more evidence is put by adding sentence and reference in line 55-57
- Line 83: what is the purpose of diluting extracted viral RNA 1:10?
Re: sorry for not having been clear. The RNA dilution is used for any potential dilution effect against inhibitory substances contained in the RNA sample tested. An explanation has been added in line 88
- Line 101: diluted sample or diluted extracted RNA?
Re: diluted extracted RNA. Change applied in line 108
- Line 138 – 139, consider rephrasing ‘RNA viral samples’ to ‘viral isolates’.
Re: thanks for your advice. Sentences rephrased in lines 153, 155, 204
- Smith DB et al have recently published updated reference sequences for HEV-A (PMID: 32469300). Major changes are required to the reference sequences used in the manuscript. Reference 27 should be updated to this reference.
Re: Thank you for this very important consideration. Smith et al. (2020) WGS references have been included in our analysis. In particular, the updated tree has been built by using a total of 82 WGS, against n.57 used in the previous draft version of the paper.
According to the other reviewer’s comments, the supplementary table has been removed so no ref updates are anymore needed.
- The reference mentioned in point 5 has extensively updated the landscape of genotype 3 and modified some of the rules of subtype assignation. JQ953664 is now reassigned as subtype 3l (not 3h as mentioned in supp. table 1). In fact, there is even a new subtype 3m. The authors will need to reanalyze their sequence ‘l-p’ to see whether it really should be assigned to a new subtype. I think the 3c and 3e assignations of the remaining boar isolates are accurate, but the authors must rigorously reanalyze the data using the most updated reference sequences.
- The rationale for assigning ‘3l-p’ to a new subtype is unclear.
Re to pint 6 and 7: A new Figure 1 has been produced. We confirmed the 3c and 3e assignations previously reported. The HEV3 “l-p” sequence now falls into the HEV3-f cluster but the author think this sequence will need much in depth investigation as reported in results (lines 167-168) and discussion (lines 232-240) sections.
- Line 145-146: ‘medium’ = mean?
Re: word changed in lines 160-161
- Isolates obtained in this study should be clearly marked in figure 1.
Re: Thank you for the advice. The sequences from this work are now marked by the use of black circles
- WGS of wild boar could be compared to Italian human and swine 3e sequences to improve the relevance of this study to public health.
Re: the authors share the opinion of this valuable reviewer about the importance of enforcing the result analysis with more Italian WGS from humans and pigs. Unfortunately, there is lack of these WGS data availability. We better explained the concept in line 257-260
Reviewer 2 Report
The authors describe the results of a HEV screening exercise among wild boars in a region of Italy with high human hepatitis E prevalence. They find evidence of circulation of HEV-3c, 3e and what they describe to be a putative novel HEV-3 subtype. It is good that they do WGS for accurate genotyping.
The main problem is that Smith DB et al have just updated reference sequences for HEV in July 2020 (PMID: 32469300) and they have already assigned sequences to new HEV subtypes 3l and 3m. The rules for assigning new subtypes have been clearly described. The authors will need to update some of their WGS reference sequences and re-run the analysis with respect to this new publication. As it stands, their conclusion that they have a novel subtype has not been confirmed.
- Is consumption of wild boar meat common in the Abruzzo region?
- Line 83: what is the purpose of diluting extracted viral RNA 1:10?
- Line 101: diluted sample or diluted extracted RNA?
- Line 138 – 139, consider rephrasing ‘RNA viral samples’ to ‘viral isolates’.
- Smith DB et al have recently published updated reference sequences for HEV-A (PMID: 32469300). Major changes are required to the reference sequences used in the manuscript. Reference 27 should be updated to this reference.
- The reference mentioned in point 4 has extensively updated the landscape of genotype 3 and modified some of the rules of subtype assignation. JQ953664 is now reassigned as subtype 3l (not 3h as mentioned in supp. table 1). In fact, there is even a new subtype 3m. The authors will need to reanalyze their sequence ‘l-p’ to see whether it really should be assigned to a new subtype. I think the 3c and 3e assignations of the remaining boar isolates are accurate, but the authors must rigorously reanalyze the data using the most updated reference sequences.
- The rationale for assigning ‘3l-p’ to a new subtype is unclear.
- Line 145-146: ‘medium’ = mean?
- Isolates obtained in this study should be clearly marked in figure 1.
- WGS of wild boar could be compared to Italian human and swine 3e sequences to improve the relevance of this study to public health.
Author Response

(The authors gave the same response as above.)

Round 2
Reviewer 1 Report
I thank the authors for having address my concerns and modified the manuscript accordingly.
Figure 1: it is surprising that one sequence was isolated in human/pig of different countries. For instance, JQ13794 and other. Please check in Genbank the host and country because a sequence is usually unique and correspond to one host from one country.
Reviewer 2 Report
The manuscript has been improved.